# In Vitro Cytotoxic Activity of Methanol Extracts of Selected Medicinal Plants Traditionally Used in Mexico against Human Hepatocellular Carcinoma

**DOI:** 10.3390/plants11212862

**Published:** 2022-10-27

**Authors:** Joel H. Elizondo-Luévano, Ricardo Gomez-Flores, María J. Verde-Star, Patricia Tamez-Guerra, César I. Romo-Sáenz, Abelardo Chávez-Montes, Nancy E. Rodríguez-Garza, Ramiro Quintanilla-Licea

**Affiliations:** 1Departamento de Química, Facultad de Ciencias Biológicas (FCB), Universidad Autónoma de Nuevo León (UANL), San Nicolás de los Garza, N.L., Monterrey 66455, Mexico; 2Departamento de Microbiología e Inmunología, Facultad de Ciencias Biológicas (FCB), Universidad Autónoma de Nuevo León (UANL), San Nicolás de los Garza, N.L., Monterrey 66455, Mexico

**Keywords:** anticancer, antioxidant activity, cytotoxic activity, hemolytic activity, medicinal plants, natural products, phytochemicals, plant extracts, Mexican plants, Mexican ethnobotany

## Abstract

Medicinal plants are traditionally used in Mexico to treat diseases such as cancer. The present study aimed to evaluate the cytotoxic, antioxidant, and anti-hemolytic activity of 15 plants of ethnopharmacological use in Mexico. For this, plant methanol extracts were prepared by the Soxhlet method, after which their cytotoxic activity was evaluated against human hepatocellular carcinoma (HEP-G2) and monkey kidney epithelial (Vero) cells by the 3-(4,5-dimethylthiazol-2-yl)-2,5-diphenyltetrazolium bromide (MTT) reduction colorimetric assay. The selectivity index (SI) of each extract was then determined by the IC_50_ ratio of normal to tumor cells. We showed that *Ruta chalepensis* extract possessed an IC_50_ of 1.79 µg/mL and 522.08 µg/mL against HEP-G2 and Vero cells, respectively, resulting in an SI of 291.50. Furthermore, antioxidant activity was evaluated by the 1,1-diphenyl-2-picrylhydrazyl (DPPH) radical scavenging technique, where the best antioxidant potential was shown by the *Heterotheca inuloides* extract (IC_50_ = 19.24 µg/mL). Furthermore, the hemolytic potential was determined against human erythrocytes, which showed that the extracts with the highest anti-hemolytic activity were *Smilax aspera* (IC_50_ = 4.41 µg/mL) and *Amphipterygium adstringens* (IC_50_ = 5.35 µg/mL). In conclusion, we observed that *R. chalepensis* methanol extract possesses cytotoxic activity against HEP-G2 cells, without affecting non-tumorigenic Vero cells. Our results indicated the antitumor potential of medicinal plants used in Mexico.

## 1. Introduction

According to the World Health Organization, cancer is one of the leading causes of death worldwide, with 19.3 million cases and 10 million deaths in 2020 [1]. Among the most critical cancers, hepatocellular carcinoma (HCC) is the most frequent primary liver neoplasm globally, requiring a multidisciplinary approach for diagnosis, treatment, and prevention. This pathology is one of the major causes of cancer-related deaths globally [2]. HCC prognosis is complex because of the competing risk imposed by underlying cirrhosis and the presence of malignancy. For this reason, there is a constant search for better diagnostic, screening, and therapy strategies to improve the prognosis and treatment of this neoplasm [3]. Identification of serum biomarkers is currently used in predictive HCC, since some of them are essential as screening tools for therapy selection, surveillance, diagnosis, prognosis, and post-treatment evaluation [4].

Ultrasound surveillance allows diagnosis at early stages, when the tumor may be curable by resection, liver transplantation, or ablation, and a 5-year survival of more than 50% can be achieved.[5]. Liver transplantation may be the most useful for people who are not candidates for resection. However, shortage of donors limits its applicability [6]. In addition, the extreme toxicity of most anticancer drugs prompts the search for new drugs that may prevent or slow cancer growth with less toxicity and more safety [7]. Therefore, developing new, safe, and more specific biological targets is essential, especially for the most aggressive tumors [1].

Humans have been using plants as a source of medicine since ancient times, and several modern drugs are derived from medicinal plants [8,9]. Alternative drugs are conventionally applied in many types of cancers to reduce chemotherapy’s toxicity and adverse effects. A variety of phytonutrients found in fruits, herbs, and spices act as potent cancer prevention agents by preventing the overproduction of toxic chemicals in the body [7]. It is well recognized that more than 70% of anticancer drugs have a natural origin, and more than 3000 plants with antitumor properties have been widely studied [10]. The scientific community has shown that cancer patients often use alternative therapies, particularly medicinal plants, which have served as the basis for discovering new anticancer drugs [11]. Among the most significant examples of plant-derived anticancer agents are vincristine, which has activity against leukemia, Hodgkin’s and non-Hodgkin’s lymphoma, and solid cancers such as breast and lung cancer, vinblastine with activity against breast, renal, and lymphoma cancer, docetaxel against breast and lung cancer, and paclitaxel against ovarian, breast, lung, and bladder cancer [12].

Plants are a source of bioactive molecules with antibacterial, antiparasitic, healing, analgesic, anti-inflammatory, and tissue repairing properties, representing a potential alternative in cancer treatment [13,14,15]. Approximately 80% of developing countries’ world population adopts plants to treat various ailments [16]. Mexico has used medicinal plants since pre-Hispanic times and is considered one of the leading countries with broad plant diversity. About 30% of Mexican patients diagnosed with cancer use plants as complementary medicine [17].

In the last two decades, research concerning natural medicinal products has increased due to the discovery of new molecules with pharmaceutical interest based on ethnopharmacological knowledge. However, more than 90% of plant species have not yet been studied [18]. The present study aimed to evaluate the in vitro antitumor, antioxidant, and anti-hemolytic activities of methanol extracts of *Amphipterygium adstringens*, *Argemone mexicana*, *Artemisia ludoviciana*, *Cymbopogon citratus*, *Heterotheca inuloides*, *Jatropha dioica*, *Justicia spicigera*, *Larrea tridentata*, *Mimosa tenuiflora*, *Psacalium peltatum*, *Pseudognaphalium obtusifolium*, *Ruta chalepensis*, *Semialarium mexicanum*, *Smilax aspera*, and *Tagetes lucida*. These plants were selected based on ethnomedical reports showing their use in different regions of Mexico, particularly against cancer [10].

*A. adstringens* possesses cytotoxic activity against prostate adenocarcinoma (PC-3), ovarian adenocarcinoma (OVCAR-3), and colon adenocarcinoma (HT-29) cell lines, whereas *A. mexicana* is used to treat stomach problems, intestinal infections, and parasites [19], and in vitro, it inhibits lung (A549), colon (HT-2), and human larynx (HLaC) cell lines growth [20]. In addition, *A. ludoviciana* is used in combination with antibiotics to treat tuberculosis [21] but has not been reported against cancer. Furthermore, *C. citratus* essential oil has shown cytotoxic activity against HT-29 and Caco-2 human colorectal adenocarcinoma cell lines [22], whereas *H. inuloides* has traditionally been used to treat a wide range of diseases in Mexico, such as rheumatism, inflammation, and muscle pain, as well as dental diseases and gastrointestinal disorders [23]. *J. dioica* extracts have been evaluated in vitro in mouse peripheral blood cells, mouse fibroblasts, human hepatocytes, and pig epithelial cells with low or nil toxicity [24] but their antitumor potential has not yet been investigated. In addition, *J. spicigera* is traditionally used in Mexico to treat dysentery, diabetes, leukemia, and anemia, and its ethanol extract, as well as that of *T. lucida*, showed cytotoxic effect against human adenocarcinoma HeLa cells [25], whereas *L. tridentata* is recognized for its various traditional uses [26]. Nordihydroguaiaretic acid, one of *L. tridentata* most studied secondary metabolites, has shown biological activity against reactive oxygen species, activation of the endogenous antioxidant response mediated by nuclear factor erythroid 2, and inhibition of lipoxygenases [27].

In addition, *M. tenuiflora* is popularly used because of its antimicrobial and antifungal properties due to its flavonoids and tannins [28]. Plants of the *Asteraceae* family, such as *P. peltatum* and *P. obtusifolium,* possess sesquiterpenes such as cacalol and cacalone that increase insulin levels and keep hypoglycemic, anti-inflammatory, and antioxidant activities [29]. *R. chalepensis* is one of the most widely used plants in Mexico and worldwide, whose phytochemical compounds have shown antiparasitic, antifungal, antioxidant, antibacterial, and cytotoxic activities [30]. In addition, the root bark of *S. mexicanum* was reported to possess a cytotoxic effect against the human breast cancer cell line MCF-7, with promising results [31]. Studies on the chemical components of *Smilax* species such as *S. aspera* have revealed the presence of steroidal saponins, flavonoids, phenylpropanoids, and stilbenoids, with antifungal, anti-inflammatory, and cytotoxic activities against different cell lines [32].

In this investigation, we evaluated the cytotoxic action of plant extracts against the human HCC cell line HEP-G2, as compared with that on normal monkey kidney epithelial cells (Vero cells) to determine cancer selectivity. HCC HEP-G2 cells are widely used in cytotoxicity studies because they are easy to maintain in culture [33] and they preserve hepatic functions similar to those of primary human hepatocytes [34]. They are useful to evaluate regulation of enzymes that metabolize pharmacological agents [35]. One of the advantages of using the Vero cell line is that it indefinitely multiplies, whereas primary liver cells have a limited lifespan and die after a number of generations; therefore, the Vero cell line is easy to handle and represents an unlimited source of self-replication with a relatively high degree of homogeneity [36].

Therefore, based on epidemiological data and empirical knowledge, these medicinal plants, commonly used by ethnic groups in Mexico, were studied to validate their therapeutic properties. This investigation contributes to the knowledge and dissemination of traditional Mexican medicine and provides information for further studies of bioactive molecules present in plants that may be useful in treating human hepatocarcinoma.

## 2. Results

### 2.1. Plant Identification

Table 1 shows plants used in the present study, which were identified by the FCB taxonomist with their corresponding identification voucher numbers, the family to which they belong, their common name, the parts used to perform the extractions, and the extraction yields obtained for each extract. These plants were selected based on empirical or scientific knowledge of their use in Mexico. The plant with the highest extraction yield was *A. adstringens*, with 38.36%, and the plant with the lowest extraction yield was *S. mexicanum*, with an extraction percentage of 10.52.

### 2.2. Cytotoxic Activity

Table 2 shows the cytotoxicity results of the extracts against tumor and normal cells and the SI obtained for each extract. HEP-G2 cells were compared against Vero cells since these are adherent cells. The extracts with the highest cytotoxic activity against the HEP-G2 cell line were those of *J. spicigera* (IC_50_ = 2.92 µg/mL) and *R. chalepensis* (IC_50_ = 1.79 µg/mL), which statistically presented similar activity, followed by *J. dioica* (IC_50_ = 12.34 µg/mL) and *A. adstringens* (IC_50_ = 41.77 µg/mL). The other extracts presented IC_50_ greater than 350 µg/mL against HEP-G2 cells. Vincristine positive control showed cytotoxicity percentages of 11.27 and 19.65 against Vero and HEP-G2 cell lines, respectively, after 48 h incubation. Extracts with SIs higher than three were classified as promising. This value indicates that the extract is three times more cytotoxic for the tumor cell line, compared with the normal cell line [37]. The highest SI’s against HEP-G2 cells corresponded to *R. chalepensis*, *J. spicigera*, *J. dioica*, and *A. adstringens* extracts, with an SI of 291.50, 18.84, 5.72, and 4.74, respectively. *R. chalepensis* extract showed the highest SI. The selectivity behavior of extracts provides information on being selectively cytotoxic to cancer cells and less cytotoxic to normal cells. Extracts with this behavior will be considered suitable for the bidirected isolation of their phytocompounds.

### 2.3. Antioxidant Activity

The extract with the highest antioxidant activity was *H. inuloides*, with an IC_50_ = 19.24 µg/mL. The other extracts showed IC_50_ above 500 µg/mL (Table 3). The extract with the lowest antioxidant activity was *S. mexicanum*, with an IC_50_ = 1639.79 µg/mL. However, none of the extracts was significantly better than the positive control.

### 2.4. Hemolytic and Anti-Hemolytic Activity

The in vitro hemolytic test was commonly used in the pharmaceutical industry to screen therapies throughout the early stages of clinical development [38]. In the tests of the hemolytic activity of the methanol extracts against human erythrocytes, it was found that the extract with the highest hemolytic activity was *A. adstringens* (IC_50_ = 203.62 µg/mL), followed by *A. adstringens* (IC_50_ = 545.74 µg/mL). However, the other extracts resulted in IC_50_ above 600 µg/mL. Specifically, the extracts corresponding to *J. spicigera*, *M. tenuiflora*, *P. peltatum*, and *P. obtusifolium* were the least hemolytic with IC_50_ above 2500 µg/mL (Table 4).

The AAPH radical-induced hemolysis protection assay, which causes lipid peroxidation of normal red blood cells [39], and is considered an ex vivo model to demonstrate the high antioxidant potential of some plant species [40]. When the anti-hemolytic activity was evaluated, it was found that the extracts with the highest anti-hemolytic activity were *S. aspera* (IC_50_ = 4.41 µg/mL), *A. adstringens* (IC_50_ = 5.35 µg/mL), *H. inuloides* (IC_50_ = 5.42 µg/mL), and *P. peltatum* (IC_50_ = 5.92 µg/mL), which statistically behaved in the same manner (Table 4). The extract with the lowest anti-hemolytic activity was *L. tridentata* (IC_50_ = 777.85 µg/mL).

## 3. Discussion

Cancer is the second leading cause of death in Mexico. Most cases are detected in advanced stages, and the Mexican population commonly applies natural therapies. More than 300 complementary and alternative medicine treatments are currently used in Mexican folklore [17]. Some studies have shown that plants of the *Anacardiaceae* [41]*, Papaveraceae* [42]*, Compositae* [43]*, Euphorbiaceae* [44]*, Acanthaceae* [45], and *Rutaceae* [30] families have antioxidant, anti-inflammatory, antiparasitic, antimicrobial, and anticancer properties [46,47,48]. The search for new compounds with antitumor potential represents a priority in the pharmaceutical area due to new variants resistant to conventional antineoplasic drugs [10].

Medicinal plants produce bioactive compounds with various biological functions that possess a high content of flavonoids, coumarins, carbohydrates, tannins, sterols, quinones, and alkaloids, as observed in *R. chalepensis*, *J. spicigera*, *J. dioica,* and *A. adstringens*, which showed high selectivity against tumor cells. These compounds may be an effective alternative for new anticancer molecules [49]. In our study, the species showing in vitro antineoplasic effects against HEP-G2 cells were *R. chalepensis* (SI = 291.50), *J. spicigera* (SI = 18.84), *J. dioica* (SI = 5.72)*,* and *A. adstringens* (SI = 4.74). When the SI value of plant extracts is greater than two, it is accepted that those have a selective toxic effect on cancer cells; on the other hand, an SI value of less than two shows low toxicity on cancer cells [50]; therefore, plant extracts with an SI less than two were considered to have no cytotoxic activity against HEP-G2 cells.

Our study demonstrated that *R. chalepensis* was cytotoxic against the human tumor cell line HEP-G2 and showed the highest SI compared with the other evaluated plants (Table 2). Alkaloids and furanocoumarins are among the primary metabolites identified in plants from the *Rutaceae* family [51,52]. These metabolites may be responsible for the cytotoxic activity we observed, since several studies evidenced the cytotoxic effect of *R. chalepensis* extracts against HeLa, MCF-7, and human squamous cell carcinoma (A431) cell lines were due to furanoacridones and acridone alkaloids (arborinin and evoxanthin) [53,54]. *R. chalepensis* has also been shown to possess antioxidant and hypoglycemic properties, protecting against lipid peroxidation through inhibiting α-amylase and α-glucosidase enzymes [52,55].

*J. spicigera* showed an SI index of 18.84 when evaluated in Vero and HEP-G2 cells, since it did not show significant toxicity in Vero cells (IC_50_ = 54.91 μg/mL) compared with HEP-G2 cells (IC_50_ = 2.92 μg/mL). Thus, we may consider it an excellent candidate to investigate against other tumor cell lines. Its cytotoxic activity on human breast cancer cell lines T47D, MCF-7, and HeLa cells with IC_50_ < 30 μg/mL has been previously reported [25,56]. Glycosides of kaempferol, such as kaempferitrin, were isolated from this plant [57,58] and identified as the primary metabolite that exerted anti-oxidant, anti-inflammatory, proapoptotic, cardio-protective, and anti-cancer activities [59].

The root of *J. dioica* is popular in Mexican herbal medicine to treat various diseases. It has also shown cytotoxic, antimicrobial, and antiviral activities. Components found in *J. dioica* are mainly diterpenes, such as citlalitrione, riolozane, 6-epiriolozatrione, citlalitrione, and jatrophatrione, and riolozatrione [60,61,62]. When we evaluated the cytotoxic activity of *J. dioica*, we found that the methanol extract has potent activity against HEP-G2 (IC_50_ = 12.34 μg/mL) cells. We also determined that the extract of *J. dioica* showed an SI of 5.72 (Table 2). Diterpenes are probably responsible for the cytotoxic activity since abietane diterpenoids have demonstrated in vitro cytotoxic activity against HL-60 and A549 cell lines and high antioxidant activity against Trolox [63]. In addition to the cytotoxic activity of the diterpene lriolozatrione, it possesses antiviral activity against HSV-1 and HSV-2 [64]. We also determined its antihemolytic potential against the AAPH radical, making it a plant of interest for future studies.

*A. adstringens* extract showed cytotoxic activity against HEP-G2 (IC_50_ = 41.77 µg/mL) with an SI of 4.74 against Vero cells. *A. adstringens* is traditionally used to treat gastric ulcers and cancer. This plant is rich in anacardic acids, 6-pentadecyl acid being the most abundant, which has been determined to be cytotoxic against the human colorectal adenocarcinoma Caco-2 [65], the human gastric adenocarcinoma AGS, and the chronic myeloid leukemia K562 [66], without affecting human PBMC [66]. In another study, the antiproliferative effect of the methanol extract of *A. adstringens* bark against the human ovarian adenocarcinoma OVCAR-3, the melanoma UACC-62, the colorectal adenocarcinoma HT-29, the prostate adenocarcinoma PC-3, the glioblastoma multiforme U251, the lung carcinoma NCI-H460, and the renal adenocarcinoma 786-O was determined, which ranged from 4.4 µg/mL to 28.0 µg/mL [67]. Such results agreed with our study, indicating the potential activity of *A. adstringens* extract against different cancer cells and showing high anti-hemolytic activity. Hence, it is essential to continue with studies of this plant, including its mechanisms. Furthermore, *A. mexicana* antitumor activity against MCF-7 and MDA-MB-231 breast cancer cell lines through the induction of the Estrogen receptor alpha (Erα) gene has been previously reported [68]. This plant is rich in alkaloids, which have significantly reduced cytotoxicity to the human colorectal adenocarcinoma cell line SW480 [69]. The other methanol extracts had cytotoxic activity against Vero cells without significantly affecting HEP-G2 cells. These extracts had SIs from 0.02 to 1.48. However, some of these plants have been ethnopharmacologically used to treat other types of cancer cell lines, such as HT29, K-562, HCT-15, SW-480, and MCF-7 cells [10,11]. However, our results showed that those plants did not have cytotoxic activity against HEP-G2 cells.

Regarding the antioxidant activity related to the DPPH free radical scavenging effect, the samples with an IC_50_ ≤ 50 µg/mL were considered to have a relevant antioxidant activity [70]. Some natural components such as phenols, alkaloids, and terpenoids in medicinal plants possess antioxidant activity and are considered necessary because of their numerous health benefits, as these molecules can scavenge free radicals [40]. Although the in vitro evaluation of free radicals is useful to demonstrate the antioxidant potential of a sample, we suggest the need to show the effects in a more complex physiological system to confirm the antioxidant potential of natural extracts in an ex vivo model, such as the assay for protection against AAPH radical-induced hemolysis in human erythrocytes [70,71].

Only *H. inuloides* extract showed significant antioxidant and anti-hemolytic activity compared with the other evaluated extracts, showing IC_50_ of 19.24 μg/mL and 5.42 μg/mL, respectively. These data agree with a study conducted in 2010, showing the antioxidant activity of its methanol extract [72]. However, this extract did not have a similar effect than the positive control, which had an IC_50_ = 7.23 μg/mL. The other extracts showed IC_50_ > than 500 µg/mL. Therefore, they were not considered to have high antioxidant activity. *H. inuloides* has been traditionally used to treat a wide range of diseases in Mexico, such as rheumatism, topical skin inflammation, and muscle pain, among other conditions associated with inflammatory processes or pain. Reports regarding the toxicity of *H. inuloides* are limited to date [23]. On the other hand, we found anti-hemolytic activity and hemolytic potential with IC_50_ of 28.29 μg/mL and 738.73 μg/mL, respectively, for *R. chalepiensis*. However, we did not find an antioxidant activity as effective as ascorbic acid or *H. inuloides* extract.

We observed the antioxidant activity of *J. spicigera* extract by the DPPH method with an IC_50_ of 924.92 μg/mL, whereas Baqueiro-Peña in 2017 [45] determined an IC_50_ of 880.00 μg/mL. We found only one report regarding the evaluation of *J. spicigera* extracts on human erythrocytes by the AAPH method, where they reported an antihemolytic activity with an IC_50_ = 13.5 µg/mL. However, they performed an extraction with a solvent mixture of dichloromethane:methanol (1:1), which may have resulted in a different outcome than ours. Therefore, this is the first report regarding the antihemolytic activity of *J. spicigera* methanol extract with an IC_50_ of 81.08 μg/mL. In addition, they reported antioxidant activity with IC_50_ = 35.8 µg/mL, as determined by the DPPH method.

The antioxidant (IC_50_ = 312.50 μg/mL) and anti-hemolytic activity of *A. mexicana* methanol extract at 400 μg/mL has been reported [20]. However, berberine, one of its main components, was observed to have an antioxidant effect with IC_50_ = 168.18 μg/mL but had a deficient anti-hemolytic activity, which decreased as the concentration increased [47]. In our study, we determined that *A. mexicana* extract possesses DPPH uptake potential with IC_50_ = 565.98 μg/mL and a low hemolytic activity (IC_50_ = 973.88 μg/mL). This difference may be due to the concentration of secondary metabolites in the extract or by the area and date of collection of the plant since these factors may cause diversity in the concentration of metabolites in the plant [73].

The inhibitory activity of the extracts against AAPH radicals was related to the most effective extracts of *A. adstringens* (IC_50_ = 5.35 ± 1.63 µg/mL), *H. inuloides* (IC_50_ = 5.42 ± 0.89 µg/mL), *P. peltatum* (IC_50_ =5.92 ± 1.29 µg/mL), and *S. aspera* (IC_50_ = 4.41 ± 0.69 µg/mL) since they did not present statistically significant differences. The extract of *L. tridentata* (IC_50_ = 777.85 ± 18.58 µg/mL) showed the lowest antihemolytic activity. The protective role against hemolysis by AAPH has been attributed to the polyphenol content of plants because they interact with erythrocyte membrane components by preventing oxidation of membrane proteins and lipids via hydrogen bonds [71].

## 4. Materials and Methods

### 4.1. Chemicals and Reagents

All chemicals and solvents were of analytical grade. Dulbecco’s modified Eagle medium (DMEM), 1% antibiotic/antimycotic solution, and fetal bovine serum (FBS) (Gibco BRL, Grand Island, NY, USA) were used. Dimethyl sulfoxide (DMSO), 2,2′-azobis (2-methylpropionamidine) dihydrochloride (AAPH), 2,2-diphenyl-1-picrylhydrazyl (DPPH), methyl alcohol (MeOH), 3-(4,5-dimethylthiazol-2-yl)-2,5-diphenyltetrazolium bromide (MTT), ferric chloride, sodium hydroxide (NaOH), and sulfuric acid were obtained from Sigma-Aldrich (St. Louis, MO, USA). Vincristine sulfate (VS) salt was purchased from Hospira Inc. (Lake Forest, IL, USA).

### 4.2. Plant Material and Extraction

Plants used in this study were identified by Professor Dr. Marco A. Guzman Lucio, Chief of the Herbarium of FCB at UANL, Nuevo León, México, who was assigned a voucher number to each specimen (Table 1). Specimens were deposited in the FCB-UANL herbarium. The names and botanical families of species were taxonomically validated using WorldFlora Online, http://www.worldfloraonline.org (accessed on 1 August 2022). Plant extraction was performed by subjecting 60 g of milled dry plant material to 600 mL of absolute MeOH in Soxhlet equipment for 48 h [51]. Extracts were then filtered, rotaevaporated, and stored at 4 °C in amber bottles until use. The following Formula (1) was used to calculate the extraction yield percentage [13]:(1)$$\%\;\mathrm{Yield}=\frac{\mathrm{Final}\;\mathrm{weight}}{\mathrm{Initial}\;\mathrm{weight}}\times 100$$

### 4.3. Tumor and Normal Cells

Human hepatocellular carcinoma (HEP-G2; ATCC HB-8065) and monkey kidney epithelial cells (Vero; ATCC CCL-81) were grown in DMEM supplemented with 10% heat-inactivated FBS and 1% antibiotic/antifungal solution (referred as a complete DMEM culture medium) [74]. Cells were cultured at 37 °C in an atmosphere of 5% CO_2_ and 95% air. Before evaluating the extracts, cells were incubated for 24 h for adaptation. To test the cytotoxicity of plants, cells were treated with methanol extracts at concentrations ranging from 31.25 µg/mL to 1000 µg/mL of each extract in a final 200 µL for 48 h. The positive control consisted of 0.05 µg/mL of VS, whereas the negative control was culture medium alone. A colorimetric assay determined cell viability by adding 15 µL of MTT/well (final concentration of 500 µg/mL) and incubating for three hours. Formazan crystals were dissolved with 80 µL/well of DMSO, and optical densities (OD) were measured at 570 nm on a MULTISKAN GO microplate reader (Thermo Fisher Scientific, Waltham, MA, USA). Cell viability was determined using the following Formula (2):(2)$$\%\;\mathrm{Cell}\;\mathrm{viability}=\frac{\mathrm{OD}\;\mathrm{Treatment}}{\mathrm{OD}\;\mathrm{Negative}\;\mathrm{control}}\times 100$$

Finally, the selectivity index (SI) was determined for each extract evaluated. SI are used to assess cytotoxic potential relative to toxicity to normal cells, where high SI indicates high potency and low cellular toxicity [75]. They were obtained for each extract evaluated after dividing the IC_50_ (half maximal inhibitory concentration) by the specific IC_50_ against HEP-G2 tumor cells after 72 h incubation [76]. According to Bezivin et al. (2003), the selectivity index is interesting in the case of values greater than three [37]. Thus, in our study, any sample that has an SI value greater than three was considered to have high selectivity. SI was calculated using the following Formula (3):(3)$$\mathrm{SI}=\frac{{\mathrm{IC}}_{50}\;\mathrm{Normal}\;\mathrm{Cells}\;\mathrm{Value}}{{\mathrm{IC}}_{50}\;\mathrm{Tumor}\;\mathrm{Cells}\;\mathrm{Value}}$$

### 4.4. Antioxidant Activity

The antioxidant potential of 31.25 µg/mL to 500 µg/mL plant extracts was tested by the DPPH radical scavenging assay [77]. For this, we incubated 100 μL of 125 μM DPPH to 100 μL of each sample for 30 min at 37 °C in darkness. ODs at 517 nm were then measured in a spectrophotometer Genesys 20 (Thermo-Fisher Scientific, Waltham, MA, USA), using 50 µM ascorbic acid as a positive control (C+) and MeOH as a blank. The reduction percentage was calculated using the following Formula (4):(4)$$\%\;\mathrm{DPPH}\;\mathrm{reduction}=\frac{\mathrm{OD}\;\mathrm{Treatment}}{\mathrm{OD}\;\mathrm{Blank}}\times 100$$

IC_50_ values were calculated as the concentration of sample required to scavenge 50% of DPPH.

### 4.5. Hemolytic Activity Test

The in vitro hemoglobin denaturation and hemolysis test is recommended and used for a battery of tests aimed at checking the toxicity of different products in the pharmaceutical industry, which are useful to examine therapies throughout the early stages of clinical development [78]. The hemolytic activity of plant extracts was evaluated by the hemolysis test [79]. Ten milliliters of human blood from healthy donors was collected by venipuncture in tubes with anticoagulant (EDTA), after which blood was centrifuged to separate erythrocytes from plasma at 1500 rpm (15 min at 25 °C) and washed three times with 10 mL of PBS (pH 7.4). Erythrocytes were then suspended at 5% v/v in PBS for the tests. For the hemolysis assay, the erythrocytes suspension was mixed with the extracts dissolved in PBS at concentrations ranging from 100 μg/mL to 1000 μg/mL in 2 mL Eppendorf (Eppendorf^®^ AG, Hamburg, Germany) microcentrifuge tubes for 30 min at 37 °C, protected from light. After this, treatments were centrifuged at 13,000 rpm (3 min at 4 °C) and 200 μL of supernatant fluids were taken and placed on a transparent plastic 96 flat-bottomed wells microplate. The degree of hemolysis was then determined by measuring ODs at 540 nm, whereas the IC_50_ values were calculated as the concentration of sample required to hemolyze 50% of human red blood cells. The percentage of hemolysis was calculated with the following Formula (5):(5)$$\%\;\mathrm{Hemolysis}=\frac{\mathrm{OD}\;\mathrm{Treatment}}{\mathrm{OD}\;\mathrm{Positive}\;\mathrm{control}}\times 100$$

### 4.6. Anti-Hemolytic Activity Test

The inhibitory property of many antioxidants on AAPH radical-induced hemolysis has been widely studied and documented [47,77,80,81]. Hemolysis of erythrocytes is commonly used as an ex vivo model in the study of the alteration of cell membranes induced by ROS, since the thermolysis of the azoic compound AAPH generates a constant flow of peroxyl radicals at 37 °C, causing hemolysis of erythrocytes [80]. The anti-hemolytic activity was determined by the AAPH inhibition test, as previously reported [81]. For the hemolysis inhibition assay, the same steps were performed as for the hemolysis assay but in this case, the AAPH (150 mM) radical prepared in PBS was used as a hemolysis inducer (positive control). To evaluate the erythrocyte protection effect of the extracts, the previously obtained erythrocyte suspension was incubated in 2 mL Eppendorf microcentrifuge tubes with different concentrations of the extract 100 μg/mL to 1000 μg/mL plus AAPH. The mixture was set at 37 °C for 3 h at 250 rpm and protected from light. Next, the mixture was centrifuged at 13,000 rpm (5 min at 4 °C) and 200 μL of supernatant fluids were taken and placed in a transparent plastic 96 flat-bottomed wells microplate. The percentage of anti-hemolytic activity was calculated with the following Formula (6):(6)$$\%\;\mathrm{Anti}-\mathrm{hemolytic}\;\mathrm{Activity}=1-\left(\frac{\mathrm{OD}\;\mathrm{Treatment}}{\mathrm{OD}\;\mathrm{Positive}\;\mathrm{control}}\;\right)\times 100$$

### 4.7. Statistical Analysis

Statistical analyses were performed using the Graph Pad Prism 6 program (GraphPad Software Inc., San Diego, CA, USA). One-way analysis of variance was used to determine the significant difference between evaluated concentrations. The post-hoc Tukey test was used to determine the difference between treatment means. The Probit test was used to calculate the IC_50_ values. All data represent means ± SD of triplicate determinations from at least three independent experiments (*p* < 0.05).

## 5. Conclusions

The use of medicinal plants demonstrates the biotechnological potential of plants as a source of molecules with cytotoxic activity. Plant methanol extracts used in this study showed promising selectivity indexes, particularly *R. chalepensis* extract, which demonstrated its cytotoxic effect in vitro, making it a suitable candidate for future research in cancer therapies. Our study provides scientific validation of the antitumor potential of some plants commonly used in Mexico and Latin America. In addition, we showed their hemolytic or anti-hemolytic potential for the first time, suggesting that they may be promising for developing new therapies against cancer, as some of them were not toxic for red cells. Our results may contribute to the safe ethnopharmacological use of medicinal plants, which stimulates research on their bioactive compounds against cancer and the mechanisms associated with their biological activity.

## Figures and Tables

**Table 1 plants-11-02862-t001:** Plants used in this study and percentage of extraction yields.

Plant Species	Family	Mexican Common Name	Evaluated Plant Part	Voucher Number	Yield %
*Amphipterygium adstringens* (Schltdl.) Standl.	Anacardiaceae	Cuachalalate	Bark	30642	38.36
*Argemone mexicana* L.	Papaveraceae	Chicalote	Leaves	29127	11.26
*Artemisia ludoviciana* Nutt.	Compositae	Estafiate	Leaves	30643	18.39
*Cymbopogon citratus* (DC.) Stapf.	Poaceae	Zacate limón	Leaves	30644	23.04
*Heterotheca inuloides* Cass.	Compositae	Mexican arnica	Flowers	30646	20.49
*Jatropha dioica* Sessé	Euphorbiaceae	Sangre de dragón	Root	30648	15.97
*Justicia spicigera* Schltdl.	Acanthaceae	Muicle	Leaves	30649	13.18
*Larrea tridentata* (Sessé & Moc. ex DC.) Coville	Zygophyllaceae	Gobernadora	Leaves	30650	13.17
*Mimosa tenuiflora* (Willd.) Poir.	Leguminosae	Tepezcohuite	Bark	30651	10.84
*Psacalium peltatum* (Kunth) Cass.	Compositae	Matarique	Leaves	30652	10.93
*Pseudognaphalium obtusifolium* (L.) Hilliard & B.L.Burtt.	Compositae	Gordolobo	Leaves	30653	16.99
*Ruta chalepensis* L.	Rutaceae	Ruda	Root	30654	19.39
*Semialarium mexicanum* (Miers) Mennega	Celastraceae	Cancerina	Bark	30647	10.52
*Smilax aspera* L.	Smilacaceae	Zarzaparrilla	Leaves	30655	13.14
*Tagetes lucida* Cav.	Compositae	Hierbanís or Yerbaniz	Bark	30656	20.63

**Table 2 plants-11-02862-t002:** Cytotoxic activity of Mexican plant extracts against HEP-G2 tumor cells and Vero non-tumor cells.

Plant Extract	HEP-G2 IC_50_ (µg/mL)	Vero IC_50_ (µg/mL)	SI
*Amphipterygium adstringens*	41.77 ± 6.18 ^c^	197.98 ± 4.71 ^e^	4.74
*Argemone mexicana*	820.78 ± 20.81 ^h^	245.41 ± 13.05 ^g^	0.29
*Artemisia ludoviciana*	1034.76 ± 12.01 ^i,j^	197.37 ± 2.79 ^e^	0.19
*Cymbopogon citratus*	1560.01 ± 23.26 ^j^	24.47 ± 3.79 ^a^	0.02
*Heterotheca inuloides*	1002.08 ± 14.81 ^i^	36.95 ± 8.22 ^b^	0.04
*Jatropha dioica*	12.34 ± 3.12 ^b^	70.59 ± 9.73 ^d^	5.72
*Justicia spicigera*	2.92 ± 0.54 ^a^	54.91 ± 7.60 ^c^	18.84
*Larrea tridentata*	403.05 ± 13.72 ^d^	214.64 ± 1.63 ^f^	0.53
*Mimosa tenuiflora*	809.64 ± 19.72 ^h^	467.59 ± 13.72 ^h^	0.58
*Psacalium peltatum*	975.81 ± 15.19 ^i^	54.91 ± 4.94 ^c^	0.06
*Pseudognaphalium obtusifolium*	690.05 ± 10.45 ^g^	61.98 ± 2.82 ^c,d^	0.09
*Ruta chalepensis*	1.79 ± 0.38 ^a^	522.08 ± 29.96 ^i^	291.50
*Semialarium mexicanum*	527.10 ± 20.87 ^e^	20.75 ± 3.16 ^a^	0.04
*Smilax aspera*	393.05 ± 15.06 ^d^	582.11 ± 31.14 ^i^	1.48
*Tagetes lucida*	607.93 ± 11.15 ^f^	780.62 ± 9.27 ^j^	1.28

Data are means ± SD (*p* < 0.05) of the IC_50_ in µg/mL for each extract, against the evaluated cell lines. Different letters within the same column are significantly different, analyzed by the post-hoc Tukey test. The selectivity index (SI) represents IC_50_ for the normal cell line (Vero) between the IC_50_ for the cancerous cell line (HEP-G2) after 72 h. Vincristine sulfate was used as a positive control at a concentration of 0.05 µg/mL, as detailed in the Section 4.

**Table 3 plants-11-02862-t003:** Antioxidant activity of methanol plant extracts by the DPPH radical scavenging method.

Plant Extract	DPPH Assay (IC_50_ in µg/mL)
*Amphipterygium adstringens*	504.89 ± 34.33 ^c^
*Argemone mexicana*	565.98 ± 17.60 ^c^
*Artemisia ludoviciana*	723.33 ± 25.92 ^d,e^
*Cymbopogon citratus*	690.4 ± 26.37 ^d^
*Heterotheca inuloides*	19.24 ± 2.11 ^b^
*Jatropha dioica*	681.18 ± 15.64 ^d^
*Justicia spicigera*	924.92 ± 30.83 ^f^
*Larrea tridentata*	665.41 ± 31.70 ^d^
*Mimosa tenuiflora*	547.66 ± 28.87 ^c^
*Psacalium peltatum*	520.52 ± 15.69 ^c^
*Pseudognaphalium obtusifolium*	528.67 ± 25.78 ^c^
*Ruta chalepensis*	859.85 ± 25.08 ^e^
*Semialarium mexicanum*	1,639.79 ± 35.74 ^g^
*Smilax aspera*	936.5 ± 11.19 ^f^
*Tagetes lucida*	550.85 ± 16.09 ^c^
Ascorbic acid (positive control)	7.23 ± 0.03 ^a^

Data are means ± SD (*p* < 0.05) of the IC_50_ in µg/mL for each evaluated extract. Different letters within the same column are significantly different, analyzed by the post-hoc Tukey test.

**Table 4 plants-11-02862-t004:** Hemolytic and anti-hemolytic activities of Mexican plant extracts on human erythrocytes.

Plant Extract	Hemolytic Activity	Anti-Hemolytic Activity
	IC_50_ (µg/mL)	
*Amphipterygium adstringens*	203.62 ± 11.96 ^a^	5.35 ± 1.63 ^a^
*Argemone mexicana*	973.88 ± 38.46 ^f^	101.60 ± 10.21 ^d^
*Artemisia ludoviciana*	746.39 ± 12.80 ^d^	50.31 ± 7.64 ^c^
*Cymbopogon citratus*	606.82 ± 19.12 ^c^	32.01 ± 4.74 ^b^
*Heterotheca inuloides*	835.73 ± 23.73 ^e^	5.42 ± 0.89 ^a^
*Jatropha dioica*	545.74 ± 8.76 ^b^	72.92 ± 4.85 ^d^
*Justicia spicigera*	˃2500 ^†^	81.08 ± 8.10 ^d^
*Larrea tridentata*	741.71 ± 12.80 ^d^	777.85 ± 18.58 ^f^
*Mimosa tenuiflora*	˃2500 ^†^	71.24 ± 6.47 ^d^
*Psacalium peltatum*	˃2500 ^†^	5.92 ± 1.29 ^a^
*Pseudognaphalium obtusifolium*	˃2500 ^†^	143.17 ± 19.29 ^e^
*Ruta chalepensis*	738.73 ± 20.74 ^d^	28.29 ± 2.31 ^b^
*Semialarium mexicanum*	1976.75 ± 26.06 ^f^	41.32 ± 8.27 ^b c^
*Smilax aspera*	625.17 ± 13.11 ^c^	4.41 ± 0.69 ^a^
*Tagetes lucida*	843.84 ± 31.93 ^e^	62.48 ± 8.52 ^c^

Data are mean ± SD (*p* < 0.05) of the IC_50_ in µg/mL for each extract evaluated. Different letters within the same column are significantly different, analyzed by the post-hoc Tukey test. ^†^ As IC_50_ was above 2500 µg/mL, these values were not considered for the Tukey analysis.

## Data Availability

The datasets generated or analyzed during the present study are available from the corresponding author.

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
