# Peer review of "In Vitro Cytotoxic Activity of Methanol Extracts of Selected Medicinal Plants Traditionally Used in Mexico against Human Hepatocellular Carcinoma"

_plants, 2022, doi:10.3390/plants11212862_

Round 1

Reviewer 1 Report

See attachment

Reviewer 2 Report

Why are the values for Vincristine not shown in Table 3?

Why is positive control used in formula 5 and C + in formula 6?

In the discussion, the authors mention a possible telationships between the content of some groups of substances and their effects, Principal component analysis would help to reach more precise conclusions.

Reviewer 3 Report

This article presented cytotoxic activity against human cell lines of some selected plants from Mexico. This study will facilitate exploration of cytotoxic plants from the Mexcian flora. Before recommending this article for publication, there are some shortcomings for that should be resolve.

General comments

Overall, the study is well designed and presented in a good way, but mostly the literature is not cited. Also revise the grammatical mistakes and typos.

Italicize the species names.  

Abstract

The authors should highlight main findings in specific way and quantitative terms. Also add main methods used in this study.

Add units in line 26 with results.

Add conclusion in one line in the abstract.

Introduction

Add details of the hepatocellular carcinoma and its effect on different organs. Also mechanism of action on the cells.

Add significance of medicinal and traditional plants. The following studies could be cited.

https://doi.org/10.1016/j.chnaes.2021.03.009, https://doi.org/10.1016/j.chnaes.2021.08.002,

Provide details of the potential of medicinal plants against cancer.

Also specify some plants or medicines have origin from plants or plants products.

Line 48-50 cite relevant and updated literature. The following studies could help the author.

DOI: http://dx.doi.org/10.30848/PJB2022-3(19), DOI: 10.56042/ijtk.v21i3.31454,

Clarify aims and novelty of the study in the last paragraph of the introduction

Results and discussion

Discussion is well presented but some analyses are weekly compared with other studies. Also discussion of some results are missing. The authors are directed to add discussion of every section or results.

Methods

Section 4.3 write complete protocol.

Conclusion

Conclusion is well justified. 

Round 2

Reviewer 1 Report

Major criticism

Inappropriate control cell line for all experiments. Normal Monkey Kidney cells are not a control for Human liver cancer cell line HEG-2. A normal human liver epithelial cell line is strongly recommended for all studies.  Additionally, a rationale for using HEG-2 should be stated.

Minor Criticisms

In section 2.3, “SI” needs to be defined and explained.

In Section 2.5, a rationale for testing hemolytic activity is needed.
